# Utilizing Interactive Surfaces to Enhance Learning, Collaboration and Engagement: Insights from Learners’ Gaze and Speech

**DOI:** 10.3390/s20071964

**Published:** 2020-03-31

**Authors:** Kshitij Sharma, Ioannis Leftheriotis, Michail Giannakos

**Affiliations:** Department of Computer Science, Norwegian University of Science and Technology, 7491 Trondheim, Norway; midmandy@gmail.com (I.L.); michailg@ntnu.no (M.G.)

**Keywords:** eye-tracking, dual eye-tracking, multitouch interactive displays, CSCL, informal learning, collaboration outcome, mobile eye-tracking, MMLA, multimodal learning analytics

## Abstract

Interactive displays are becoming increasingly popular in informal learning environments as an educational technology for improving students’ learning and enhancing their engagement. Interactive displays have the potential to reinforce and maintain collaboration and rich-interaction with the content in a natural and engaging manner. Despite the increased prevalence of interactive displays for learning, there is limited knowledge about how students collaborate in informal settings and how their collaboration around the interactive surfaces influences their learning and engagement. We present a dual eye-tracking study, involving 36 participants, a two-staged within-group experiment was conducted following single-group time series design, involving repeated measurement of participants’ gaze, voice, game-logs and learning gain tests. Various correlation, regression and covariance analyses employed to investigate students’ collaboration, engagement and learning gains during the activity. The results show that collaboratively, pairs who have high gaze similarity have high learning outcomes. Individually, participants spending high proportions of time in acquiring the complementary information from images and textual parts of the learning material attain high learning outcomes. Moreover, the results show that the speech could be an interesting covariate while analyzing the relation between the gaze variables and the learning gains (and task-based performance). We also show that the gaze is an effective proxy to cognitive mechanisms underlying collaboration not only in formal settings but also in informal learning scenarios.

## 1. Introduction 

With the advent in interactive display technology, they have become easily accessible and as a result, their use in a variety of domains has been investigated. In the educational domain, interactive displays have been increasingly employed to support informal learning [1] in museums and as public displays. Despite their increased usage in informal learning scenarios, there is a lack of research about how to create and promote engaging, efficient and collaborative activities around interactive displays to foster learning. There have been a few studies relating the use of interactive displays in informal learning context and level of collaboration and cooperation [2]. To emphasize play, work and learning, the underlying mechanisms responsible for interaction should be adaptive to different context [1]. This is because a table top like interactive display could be considered as a ubiquitous from of learning tool [1]. There is an increasing interest in investigating the role of interactive displays to improve collaborative processes in various educational contexts, which results in a condensation of knowledge and guidelines about the usage and expectation from the interactive displays. For example, Higgins et al. [3,4] proposed an analysis of pedagogic opportunities of the interactive features of such displays using a typology of features. On the other hand, Dillenbourg and Evans [1] pointed out the risks of over-generalization and over-expectation from the interactive displays. However, past investigations are limited in terms of their scope in the sense that they were focused on the enjoyment, fun and engagement related experiences of the interactive displays [2,5]. Such studies do not focus on the relation among collaboration around interactive displays, engagement and learning gain. One such study that studied such relations did not find any significant difference for the learning gains between the conditions with the traditional methods of learning and collaborative learning supported by the interactive displays [6]. However, students who collaborated and interacted with the displays reported significant higher levels of fun and engagement [6]. The present work is motivated by such contributions and proposes further research in the direction of understanding and implementing better ways to collaboratively engage and learn with the interactive displays.

Prior studies have shown the potential of collaborative eye-tracking (also known as dual eye-tracking or DUET) in explaining the cognitive mechanisms responsible for the quality or outcome of collaboration. Previous DUET studies have shown that the eye-tracking data has the capability of explaining the task-based performance [7], collaboration quality [8], the expertise [9] and learning outcome [10]. However, to the best of our knowledge, the present study is the first one to utilize DUET in an informal learning scenario to study the collaborative eye-tracking data while combining physical and digital collaborative spaces.

Moreover, there is a recent trend shown in various learning scenarios to use the physiological data sources to explain the learning processes or to study the relation between the different variables influencing the learning gains or experiences [11,12,13,14]. In this contribution, we utilize DUET to understand how learners collaborate, engage with the content and learn with interactive displays in informal educational settings. To do so, we designed an application which enhances the capacities of the interactive display and then designed and conducted a two-staged within-group experiment. In the experiment, participants were asked to go through a set of posters and interact with the application (both in collaborative and competitive ways). We recorded the gaze of the participants while they were watching the posters and while they were interacting with the application; at the end of each session, we asked them to fill in some content specific questions and attitudinal questions. Utilizing the data from this study, we focus on how to explain the relation between the learning gains in an informal learning setting, engagement via game-play and collaboration via DUET data. More precisely, this paper contributes to the state-of-the-art in the following ways:We present the implications from the first (to the best of our knowledge) collaborative eye-tracking study in the informal learning setting.We present a method to analyze and compare the collaborative interaction of peers across different tasks (learning and gaming) and mediating interfaces (physical and digital).We provide empirical extension of the theoretical framework collaborative learning mechanisms.

## 2. Theoretical Background in Informal Learning and Collaborative Learning 

Recent computer supported collaborative learning (CSCL) research for informal learning has been focused on the process of collaborative learning rather than the outcome of the collaborative effort. For example, Tissenbaum et al. [15] proposed a framework to analyze the verbal and physical interaction of peers around an interactive table top environment and showed that having common learning goal is not a necessary condition for the success of collaborative learning sessions. Similarly, Shapiro et al. [16] proposed a new framework to visualize and analyze the interaction and movement of participants during museum visits based on the video coding. Davis et al. [17] analyzed the interaction between peers in a dyad to understand how the next step negotiation took place and how was the common understanding attained. One common theme across all these different studies is the qualitative nature of analyses. In this paper, we take a more quantitative view of the problem at hand, that is, to analyze the collaborative learning process in informal learning contexts.

One of the theoretical frameworks used to analyze the collaborative learning in informal settings is the collaborative learning mechanisms (CLM) framework [18]. The CLM framework is defined across two dimensions, explaining the different mechanisms underlying collaborative learning in informal settings. First, collaborative discussion; and second, coordination of collaboration. Collaborative discussion deals with the moments concerning the negotiation and suggestions, while the coordination of collaboration focuses on the specific moments related to the joint attention and narrations. In this way, the analysis of the collaborative sessions focuses both on the verbal and physical interactions [18]. CLM provides a way to analyze the process involving the collaborative learning, which has attracted a lot of attention in the recent CSCL research ([19,20]). 

There have been other extensions of CLM that provide in depth analysis mechanisms to study the informal collaborative learning processes, such as divergent CLM (DCLM, Tissenabum et al. [15]). DCLM framework does not necessarily assume that the goal of collaborative learning is to create a mutual understanding among the peers in a group (or dyad). However, for this contribution, we base our guiding principle on the convergent conceptual change (CCC, [21]). The CCC theory suggests that in a collaborative learning scenario the learners engage in collaborative (joint) tasks. These tasks aid in reducing the initial difference in peers’ understanding about the domain concepts. This reduction is due to the processes of argumentation, negotiation and acceptance of suggestions [22]. This convergence could also be a result of communication and the resulting socio-cognitive processes within the group that supports a collective knowledge structure, or mutually shared cognition, forming of shared mental models [23]. 

One of the common factors that most of the studies in the field of CSCL and informal learning have focused on is the analyses methods. Most of the studies mentioned in the previous sections of the exemplar studies from the theoretical contributions focused on analyzing the dialogues and gestures (manually annotated) and video analyses. This is a significant lack of automatic analyses methods that can be used to further provide feedback to the learners about their individual and/or collaborative learning processes [19]. CLM is partially based on CCC because one of the assumptions for the framework is that the main motivation of the collaborating peers is creating and sustaining mutual understanding [18]. This is quite like the scenario that we present via this contribution, where the participants must work in dyads to be able to understand a few basic neurological concepts (see Section “Research design”).

One might argue that most of the visitors in museums have diverse goals, for example, adults accompanying children might have different learning goals than those of the children themselves [24]. Thus, CCC might not be the best suited theoretical model for such “free-choice” learning contexts. There are two counter arguments for this. First, according to Dillenbourg [25] such situations are still collaborative; and second, the case with present contribution is not a typical “free-choice” environment. The main objective of the “museum-like” environment was to teach students about basic neurological concepts. Hence, there is an opportunity for having a convergent learning. Therefore, we argue that CCC and, by extension CLM is still applicable in our case.

In this contribution, we extend the CLM framework by having our focus on automatic analyses of collaborative learning in informal settings via multimodal data. We use DUET for measuring joint attention among the collaborating partners and compare the gaze similarity (a measure that captures the convergence of the eye-tracking patterns) for the pairs with the different levels of performance/learning gains. We will also use automatically extracted speech segments to compare the joint attention and other eye-tracking measurements (see Section “Measurements”) while the peers engage in dialogues. By using the automatic ways of analyzing CSCL situations, we could not only understand the process faster than the manual tagging of events, but we might also be able to predict the outcome/experiences in real-time or with fewer data points [20,26]. This opens new avenues for supporting real-time analysis of multimodal data to support CSCL. 

## 3. Related Work and Research Questions 

### 3.1. Interactive Displays in Informal Educational Settings 

With advances in interactive display technology and increased levels of affordability in recent years, there has been a growing interest in exploring their use within educational contexts. A variety of interactive displays are now available through several vendors and many learning applications have emerged. Informal learning environments, like museums, science centers and aquariums have received attention from the education research community, as supporting social interaction and collaboration is highly related with learning and engagement in informal environments [27] and interactive spaces and surfaces migrate from research lab to learning settings. Already, supporting learning with these technologies has become a vital research area in educational technology [28] and several studies have established that interactive displays have the potential to engage students in meaningful collaborative learning, especially during an informal learning visit [29,30]. 

Callahan et al. [31] identified museums, through a review of almost 150 studies, as one of the most common contexts for informal and incidental learning. Recent research shows that digital technology can provide support in creating engagement for learning in informal settings like in museums [32,33,34]. Large interactive displays have become increasingly popular in museums and other public spaces. These devices combine engaging multimedia content, sophisticated forms of interaction and new means of audience participation [17]. Interactive displays might be attractive to designers of informal learning experiences, however supporting learning and effective collaboration using such displays is challenging [35,36]. 

“Read-it” [37] was one of the first systems, combining touch technology and education, designed to scaffold reading skills development among young children (5-7 years old). Read-it was a game based application that was proven, through a pilot study, to not present any obstructions to the learning processes of the children [37]. To foster collaboration and engagement among users, Lo et al. [38] suggested using gamification elements such as badges, points (xp), achievements and expertise levels within the interactive displays. Prior studies have indicated towards a strong relation between the enjoyment and playful experiences and the students’ interaction with multi-touch technologies [39]. Another factor that has been proven to be an effective contributor to motivation and “fun” is implementing multiplayer aspects in the interactive displays. For example, Schäfer et al. [2] reported higher motivation, engagement and playfulness when the students could collaborate, cooperate and compete in a learning environment with multiple learning and playing modes. 

One educational framework that is inspired by Discovery Learning Technique proposed that the educational games if integrated with multi-touch displays in different informal learning spaces could stimulate collaboration, engagement and effective knowledge consolidation [39]. In their study based on this framework, Ardito and colleagues [39] showed that when the participants interacted with multi-touch displays they showed higher levels of engagement and collaboration as compared to the single user systems. Recently, Leftheriotis et al. [40,41] have proposed another framework to design and evaluate gamified activities in conjugation of informal learning spaces and the interactive displays. The results reported by the authors show a significant improvement in students’ enjoyment, satisfaction and knowledge gains. 

Futura [29] was a tabletop game developed to identify the key design factors while creating multi-player games with an interactive display to foster learning. Futura was reported to have high enjoyment and effectiveness by most participants [29]. One of the key reasons reported in the study was the mutual awareness of the collaborators actions in such a system [29]. Furthermore, Watson et al. [42] and Kirriemuir and McFarlane [42] call for more research around understanding the inherent capacities of interactive multi-touch displays and educational games to scaffold knowledge acquisition in students. Both contributions claim that such research is necessary for a meaningful interaction of such technology and informal education spaces.

Evans and Rick [28] present a range of exemplary projects that showcase the potential of interactive surfaces and spaces to support learning across age groups and content domains. They highlight the impetus of academic research based on such systems. In the same line of research, Martinez-Maldonado et al. [43] propose the use of learning analytics techniques to exploit further unexplored affordances of interactive surfaces that might include multi-modal analytics or analytics from heterogeneous sources of data. Even though during the last years we have seen a growing body of research on the use of interactive displays to support learning and collaboration [1,44], there is very little existing research on informal learning experiences involving rich datasets coming from learners’ interactions. Therefore, in this study we investigate the role of interactive displays in students’ collaboration, learning and engagement through the lens of data captured via learners’ gaze, interaction with the display and standardized attitudinal and performance measures. 

### 3.2. Eye-Tracking in Education 

Eye tracking has been employed to understand the learning processes and different levels of outcome in a multitude of learning scenarios. Prieto et al. [45] used eye-tracking data to explain the cognitive load that the teachers experience during different classes. In a multifaceted group of studies Prieto et al. [46] compared the “orchestration load” experienced by the teachers in various scenarios using eye-tracking data. These scenarios include different factors such as experience of the teacher, size of the class, presence of a new technology and presence of a teaching assistant. The results show that the eye-tracking data is an important source of information explaining different factors in teachers’ orchestration load and experience. 

Eye-tracking has also been used in online learning for both in individual [47,48,49] and collaborative [8] settings. On one hand, Sharma et al. [50], Van Gog et al. [49], van Gog and Scheiter [48] focus on capturing the attention of the individual learners in a video-based instructional setting to find the underlying mechanisms for positive learning outcome; on the other hand, Sharma et al. [8] focus on joint attention in remote collaborative learning scenarios to predict the learning outcome. In a co-located collaborative learning setting with tangible user interface Schneider and Blikstein [51], Schneider et al. [2,5] used eye-tracking to explain the learning processes leading to high learning outcomes. 

Recent educational research has used eye-tracking in different collaborative (remote and co-located) and individual (video-based and with cognitive-tutoring) settings and for both sides of instruction (teacher and students) in formal learning settings. In this contribution, we utilize eye-tracking capabilities to explain the relation between the learning gains in an informal learning setting, game play and collaboration. 

### 3.3. Dual Eye-Tracking for Communication and Referencing 

Prior work in the field of collaborative or Dual eye-tracking (DUET) has shown that the DUET data can help the researchers explain various cognitive processes responsible for collaboration [7,46,51,52]. A typical setup for DUET employs two eye-trackers that are synchronized using a time server. These synchronized eye-trackers collect data from the collaborating (or competing in some cases) partners while they try and solve a problem. Nüssli [52] used DUET to explain the cognitive and social dynamics between partners while they were solving various collaborative problems. Richardson and colleagues [53,54,55] proposed a concept of joint attention (or cross-recurrence) in a listening comprehension task. Cross recurrence measures the fact that how much time the partners spent looking at the same things during the similar times. In their study, Richardson and colleagues [53,54,55] reported a high correlation between cross recurrence and the comprehension of the stories. Later, in a pair programming task, Jermann and Nüssli [10] replicated these results in terms of a high correlation between collaborative program comprehension and the joint attention as measured by the cross recurrence between the pair. In a DUET experiment with Raven and Bongard problems, Nüssli and colleagues [7] showed that pairs’ gaze density and fixation dispersion were predictive of the team’s performance. Another DUET study with a collaborative version of the game “Tetris” [9] reported high prediction accuracy for the pair composition (both experts, both novices, one expert and one novice) using gaze data. 

Concerning learning gains, Sangin et al. [9] presented the students with a Knowledge Awareness Tool (KAT) summarizing and displaying the peers the expertise of their partners on a certain topic. By analyzing the DUET data from a collaborative concept map experiment Sangin et al. [56] showed that the amount of gaze on the KAT was correlated with the average learning gain of the pair. In another collaborative concept map task, Sharma et al. [50] computed the similarity between the two peers as a measure of joint attention by computing the probability of looking at the same set of objects in a given window of time. From the DUET data, Sharma et al. [50] showed that there was a high correlation between the collaboration quality and the joint attention of the pair.

In a pair-debugging task, Stein and Brennan [57] showed that if the gaze of peers is shared among the partners, it had a significantly positive effect on the debugging performance and learning of the pair. In a pair programming study Stein and Brennan [57] showed that the debugging performance of the pair was significantly better when the pairs were shown where their partner is looking as compared to when the partners had no information about each other’s gaze. In another pair programming task, Sharma et al. [47] showed that the pairs with high level of understanding had their gaze focused on the “data flow” of the program while the pairs with poor levels of understanding were reported to having read the program like one reads an English text. 

However, to the best of our knowledge, DUET has not yet been used to investigate how we learn and collaborate when we use interactive displays in informal learning contexts. Learning is enacted through multiple modalities, especially in informal contexts learners engage in significant multimodal behaviors (voice, gaze, gestures, etc.) to emphasize and de-emphasize different ideas and operate an action inside the informal context. Previous studies have focused in various modalities [58], for instance shown a strong relation between dialogues and/or speech with user’s gaze [53,54,55,59,60,61]; children responses with their gaze [21] but also actions and their gaze [56]. 

The aforementioned studies showcase the importance of DUET and how it can aid in understanding and explaining collaboration quality, expertise and joint performance. The present work utilizes the capabilities of DUET to understand and explain the collaboration and learning gains in an informal learning scenario.

### 3.4. Research Questions 

Recent studies [21] have shown the relationship between gaze (and actions) and task-based performance (and learning) in formal learning contexts. In this study, we investigate the relationship between behavioral measures and the task-based performance in an informal learning scenario. Thus, the following research questions has been devised: 

**Question 1**: How does learners’ behavior (gaze and actions) affect their performance and the learning gain? 

We designed a multi-touch game-based system to capture participants’ gaze and actions (from the game-logs) in a “museum-like” setting to study the relation between the gaze and learning gains in an informal setting. There were two versions of the game: collaborative and competitive. We are interested in how individual gaze patterns are related to the performance in the competitive version of the game; and how the dual gaze patterns are related to the collaborative version of the game. Moreover, since there are close ties between gaze and speech (or dialogues), we use the speech segments to have a deeper understanding of such relations.

In a recent study, within a formal learning context, Sharma et al. [8] showed that gaze patterns in two different sub-task have a positive relation for the students with higher learning gains. In this study, we need to identify gaze’s role in different tasks; thus, we attempt to find out: 

**Question 2**: How does the gaze change in the different tasks in informal learning context? 

In the present experiment, we are interested in examining the nature of this relation in an informal learning space. We are also interested in the face how the gaze patterns of the peers change with the presence of speech and during different key moments of the game, such as the use of power ups.

## 4. Methodology 

### 4.1. Technology

In this section, we will provide mapping of the different elements of our system and the design rationales assembled from the previous research. The details about the system, the game and the experiment will follow in the subsections. Table 1 summarizes the design rationales for the technology.

**Collaboration vs. Competition:** All the pairs played two sets of games. The collaborative version, where both the participants in a pair were supposed to provide one answer and the team got the points (xp); and the competitive version, where both participants provided answers and were scored individually. It is known in the literature that collaboration promotes synthetic skills whereas competition promotes analytic skills [63], and the combination of the two helps increase students’ motivation and their learning outcomes [64]. Competition and collaboration are both strong motivational factors [65,66], however not to the same degree for all individuals in all situations [64]. For instance, competition can be highly challenging and motivating for some, while de-motivating for others. Thus, it is important to combine them, but also identify how the two different versions might affect the participants. 

**Power Ups:** we introduced three different power ups during the game they were provided during both the collaborative as well as the competitive versions of the games. 

**Hint**: this showed the participants a clue towards the correct answer.**Double xp**: this added two times the points awarded for one question to the score of the team/player.**Pause time**: this stopped the game-play for 15 s to help give the players time to think and answers.

**Game rules**: Following were the rules of the game, they were same for the collaborative and competitive versions: for every correct answer the team/participant gets 100 xp;with every correct answer one of the three power ups increases;three power ups were: pause time double xp, and hint;for each question the team/participant got 30 s.

#### Learning Resources: Posters

To create the learning material for the experiment presented (Figure 1), we took special care for the balance between the amount of text and images, their placements and annotations of the key parts in the illustrations [67]. We present a small summary of the basic poster design principles taken care while designing the posters for this experiment.

**Spatial split-attention principle**—previous research has shown the beneficial effects of integrating pictures with explanatory text: the text that refers to the picture is typically split up in smaller segments so that the text segment that refers to a particular part of the figure can be linked to this particular part or be included in the picture (for a meta-analysis, see [68]).**Spatial segmentation of text into small paragraphs**—the explanation of different brain regions was mapped directly to their position in the anatomical brain. Prior research has shown such segmentation to be particularly useful for low prior-knowledge learners (e.g., [69,70]).**Signaling**—to help the students understand the relative positions of the different brain regions were annotated in the posters with a short description of their functionality. Research shows that signaling enhances learners’ appreciation of the learning material [71] and their learning (e.g., [72,73]).

### 4.2. Research Design 

The research design of our study is a single-group time series design [74], involving repeated measurement of a group with the experimental treatment induced between two of the measures. The single-group time series design can be diagrammed as shown below (1). As depicted, one group (G) is observed (O) and receives the treatment (X) several times.
G O1 X1 O2 X2 O3(1)

Our study consists of two treatments and three measures of learning gains (via tests), one measurement of game performance (during subjects’ interaction with the game—second treatment), as well as continuous measurement of participants’ gaze. 

To identify the baseline knowledge of the participants, they were asked to fill in a pretest about the content they were going to learn (first learning gain measurement). Then they went through five posters (first treatment) about the structure of neurons, different areas of the brain and their functions, three neurological disorders and the limbic system (two examples are shown in Figure 1). The next phase was the individual posttest (second learning gain measurement). The next phase was a gamified quiz application played in an interactive display and focusing on the same content as the posters (second treatment). The game had two modalities: in one modality, the team played collaboratively while in the second, the members of a team played against each other (the interface for the game is shown in Figure 2b). The order of the game modalities was balanced among the teams. Finally, the participants individually took a final posttest (third learning gain measurement). All the tests and the quizzes in the games were multiple-choice questions. The questions and the content for the posters were designed with the help of an expert in neuroscience at the university. The poster phase was 12–15 min long and each modality of the game (collaborative/competitive) was 6–7 min long. The gaze of the participants was recorded during both the poster and the game phase using SMI and TOBII eye-tracking glasses of 60 Hz. Figure 3 shows the schema of the study.

#### Dual Eye-Tracking

We used two eye-tracking glasses simultaneously to collect the gaze data from the dyads. These glasses are light-weight and they do not hinder the normal vision of the participants in any manner. The eye-tracking glasses have two data sources: (1) the cameras that capture the eyes of the participants and (2) the camera on the nose-bridge of the eye-tracker that records the field view of the participants. The clocks of these two cameras are the same for dual eye-tracking studies, we synchronize the two eye-trackers using a fiducial marker (some of them can be seen in the Figure 2). We ask the two participants to look at one fiducial marker together at the beginning of the experiment session and this frame is recorded in the objective cameras of both the eye-trackers. This frame is used as the starting point of the dual eye-tracking data streams from the two participants. Other than this practicality, there is no known practical concern in dual eye-tracking experiments with the mobile eye-trackers. 

### 4.3. Participants and Procedure

There were 36 university students (18 randomly formed dyads), who participated in the experiment; there were 13 females among the participants. The average age was 24.4 years (SD = 5 years). The experiment took place in a lab that was designed to simulate a museum space. Upon participants’ arrival in the lab, they filled in a pretest about the poster content; afterwards, they watched the five posters. The simple instruction for the poster phase was “go through the posters as if you were visiting a museum with your friend”. The participants could discuss the content of the posters with their partners (Figure 2a). The dyads were not mandated to stick to each other, however, most of them went through the posters together. Once they finished watching and discussing the posters, the participants individually filled the first posttest. Further, they played the gamified quiz (collaborative/competitive) where they received one of the three power ups for each correct answer: “double xp”, “pause time” and “hint” (Figure 2b). During the game phase, the participants had a maximum of 30 s to answer each question; they could go back to the posters and look for the answers. This increases the engagement during the activity and requires very good quality of collaboration, however this cost a lot of time and, as we also show from the results, it is unlikely to affect the learning gain.

Once they finished both the modalities of the game, they filled in a final second posttest. All the participants were rewarded an equivalent of $10. 

### 4.4. Measurements 

**Learning Gain:** The three variables used to measure learning gain are the scores in the pre- and posttests. We normalized test scores to be between 0 and 1. We do not consider the learning gain in this experiment, as we observed a floor effect on the pretest scores (Mean = 0.2, Median = 0.1, SD = 0.16). This shows that all the participants in the experiment had almost no prior knowledge about the content of the posters. Moreover, there is not enough variance in the pretest score of the participants for it to be a significant predictor of the posttest (first of second) scores. The posttests were developed and piloted to 4 subjects before putting them into practice. The results from the pilot indicated that subjects with no prior experience to the content area (this was our target population) could only respond to very few questions. However, after a small exposure to the content, the subjects were able to attain a better score, but it was very difficult to excel (ranging from 6–8 out of 10). Please see the Appendix A for the questions used for the pretest and the two posttests (first after the poster phase and the second after the game phase).

**Game Performance:** The participants received individual experience points (xp) during the game (both collaborative and competitive) when they replied correctly to a question. We consider the xp value as their game performance index. This was calculated using the log data from the game.

**Time on task:** This the total time the participants took to answer each question. This was calculated using the log data from the game.

**Number of “power ups” used in the game:** This is the number of different powers used (double xp, pause time and hint) by the individual players in the game. This was also calculated from the log data from the game. 

**Individual Gaze**—**AOI Transitions:** We divided the posters into different Areas of Interests (AOIs), for example, text blocks and image blocks. Next, we computed the proportions of the gaze transitions from the images in the poster to the corresponding text and the proportions of the gaze transitions from the text to the corresponding images. For example, in the Figure 1, an image to text transition would be a shift of gaze from the top left image to any of the first three paragraphs and the opposite for the text to image transitions. Any transition from the top-left image to any paragraph other than the first three ones would not be counted as a valid image to text transition. The opposite is also true for the text to image transitions. 

**Collaborative Gaze**—**Gaze Similarity:** To compute the metric for the collaborative gaze patterns, we used the same measure as used by Sharma et al. [50,59]. This measure is called the gaze-similarity and is computed as the cosine similarity of the proportionality gaze vector (Figure 4). The proportionality gaze vector is the vector denoting the proportion of the time spent by each participant looking at the different elements of the visual stimulus for a small window of time (in our case 10 s). A gaze similarity value of 1 will depict that the two peers spent the same proportions of 10 s on different AOIs. Whereas, a gaze similarity value of 0 will depict that the two peers were looking at completely different elements during a given time window of 10 s. Figure 4 shows two typical examples from similarity computation.

**Collaborative Gaze**—**Gaze Transition Similarity:** from the AOI transitions we get the AOI transition graphs (as shown in Figure 5). We compute the transition proportions for each 10 s window and compute the cosine similarities between the two transition matrices for the two participants in the pair. This similarity is called as the gaze transition similarity or transition similarity. Figure 5 shows the process of computing transition similarity and typical examples.

**Speech episodes:** during the whole interaction, we computed the amount of speech for each time window of 10 s. Further, we applied a median cut on the amount of speech to divide the segments into two categories: speech segments or no-speech segments. Finally, we apply a run length encoding to have speech episodes and no speech episodes. Figure 6 shows the schematic description of computing speech episodes.

### 4.5. Data Analysis

For this contribution, we want to emphasize on the fact that our main aim is to conduct automatic analyses of the collaborative interaction data, so that in future we could be able to provide real-time actionable feedback [19,20]. This is the reason why we are choosing the focus on the quantitative data analyses for this contribution.

To identify any order effect (to decide whether we should consider the bias from the fact that some teams played collaborative phase first and other teams played the competitive phase first) we employed different ANOVAs using the first played phase as the independent variable for all the ANOVAs. The dependent variables were the game score, the pretest score, the first posttest score and the second posttest score. 

To identify the difference between pretest, the first posttest and the last posttest we employed t-tests. To identify any relation between students pre- and post-scores, as well as the game scores we employed Pearson correlations. To investigate our research question, about the relation of learners’ behavior (gaze and actions) with their learning gain we employed Pearson correlation between AOI transitions and the pretest score, gaze similarity in the poster reading and the pretest score. 

To investigate the second research question about how gaze changes in the different tasks in informal learning we employed Pearson correlation in gaze similarity in the poster reading and playing in the interactive display. To provide the complete picture in our analysis, we also employed descriptive statistics from the game logs. These logs contain the time on task for each question and the use of power ups in the game. We used Pearson correlation to investigate the relation between the final score of the team and he number of power ups used. We also used Pearson correlation to investigate the relation between the gaze similarity during the game and the number of power ups used. We used pairwise t-tests to compare the different times the teams took to answer a specific question. 

Finally, to understand the role of the speech segments during the different phases of the experiment and the different versions of the game and their affinity with the similarities (gaze and transition) and the posttest scores (first and second), we used ANCOVA with the speech segment type (speech versus no-speech) as the covariate. 

## 5. Results 

### 5.1. Descriptive Statistics

The Table 2 summarizes the basic statistics for all the measurements used in this study. 

### 5.2. Order Effect 

We used ANOVA (the tests for the necessary conditions are shown in Table 3) to compare tests and game score across the two different starting version of the game (collaborative/competitive). We did not observe any order effect of playing collaborative/competitive versions of the game first on the game score (F (1,34) = 0.06, *p* = .68, d = 0.08). Moreover, we did not observe any order effect of playing collaborative/competitive versions of the game first on the scores from the first (F (1,34) = 1.52, *p* = .22, d = 0.41) or second (F (1,34) = 0.21, *p* = .64, d = 0.15) post-tests. This allows us to analyze the collected data without considering the bias that could have been there due to the order of the game modalities (collaborative or competitive). 

### 5.3. Learning gains 

We observe a significant average improvement from the pretest score to the first posttest score (t (69.96) = −7.91, *p* < .0001, d = 1.9, Figure 7a). The scores in the first posttest are significantly higher than the pretest scores. Even though there is improvement from the first to the second posttest, the improvement is not significant (t (69.88) = 0.39, *p* > .05, d = 0.09 Figure 7). The average improvement from the first to second posttest was 2%. There were 18 participants who did not improve, 7 participants who improved between 5% to 70%; and there were 10 participants who reduced their second posttest scores between 5% and 20%.

In terms of group performances, we observe a significant average improvement from the pretest score to the first posttest score (t (33.41) = −6.38, *p* < .0001, d = 1.5 Figure 7b). The average improvement, based on groups, from the first to second posttest was again 2%. There were 10 groups who did not improve, 4 groups who improved between 5% to 45%; and there were 4 groups who reduced their second posttest scores between 5% and 15%.

The scores in the two posttests were similar for most of the participants. We observe three significant correlations: The scores in the first and the second posttests are correlated (r (34) = 0.69, *p* < .0001). The participants who score high in the first posttest also score high in the second posttest (Figure 8a).The score in the game is correlated to the score in the first posttest (r (34) = 0.42, *p* = .01). The participants who score high in the first posttest also perform well in the game (Figure 8b).The score in the game is correlated to the score in the second posttest (r (34) = 0.34, *p* = .04). The participants who perform well in the game also score high in the second posttest (Figure 8c).

### 5.4. Poster Phase 

**AOI transitions and the first posttest score:** next, we consider the individual gaze patterns during the poster phase. We observe a significant correlation between the transitions from image to text and the first posttest score (r (34) = 0.47, *p* = .003). The participant having high proportion of the image to text transitions score high in the first posttest (Figure 9a). Moreover, we also observe a significant correlation between the transitions from the text to image and the first pretest score (r (34) = 0.48, *p* < .002). Participants that have high proportion of the text to image transitions score high in the first posttest (Figure 9b). 

**Transition similarity and the first posttest score:** we observe a significant positive correlation between the transition similarity during the poster phase and the average first posttest score (r (16) = 0.46, *p* = .05). The pairs with high transition similarity also have high average first posttest score (Figure 10a).

**Transition similarity, speech and the first posttest score****:** to study the relation between the transition similarity, speech and the first posttest score, we examined the effect of the speech segments (speech and no speech) on the relationship between the first posttest score and the transition similarity. Table 4 shows the ANCOVA results and Figure 10b shows the scatter plot. We observe two single effects and an interaction effect. Both the first posttest score and the type of speech segments are related to the transition similarity; and there is an interaction effect of these two variables on the transition similarity. The first posttest score is correlated with the transition similarity (F (1,16) = 5.01, *p* = .03, d = 1.05). We also observe that the transition similarity is significantly higher for the speech segments than the transition similarity for the no speech segments (F (1,16) = 8.42, *p* = .006, d = 1.36). Furthermore, the correlation between the transition similarity and the average first posttest score is higher for the speech segments than that for the no speech segments (F (1,16) = 4.79, *p* = .03, d = 1.03 Figure 10b).

**Gaze similarity and the first posttest score:** further, concerning the collaborative gaze patterns, we observe a significant correlation between the gaze similarity during the poster phase and the first posttest score (r (16) = 0.51, *p* = .03). The pairs having high gaze similarity have high average first posttest score (Figure 11a). 

**Gaze similarity, speech and first posttest score:** to study the relation between the gaze similarity, speech and the first posttest score, we examined the effect of the speech segments (speech and no speech) on the relationship between the first posttest score and the gaze similarity. Table 5 shows the ANCOVA results. We observe two single effects and an interaction effect. Both the first posttest score and the type of speech segments are related to the gaze similarity; and there is an interaction effect of these two variables on the gaze similarity. The first posttest score is correlated with the gaze similarity (F (1,16) = 5.74, *p* = .02, d = 1.12). We also observe that the gaze similarity is significantly higher for the speech segments than the gaze similarity for the no speech segments (F (1,16) = 15.60, *p* = .001, d = 1.86). Furthermore, the correlation between the gaze similarity and the average first posttest score is higher for the speech segments than that for the no speech segments (F (1,16) = 10.27, *p* = .003, d = 1.51, Figure 11b).

**Transition similarity and gaze similarity:** we observe a significant positive correlation between the transition similarity during the poster phase and the gaze similarity (r (16) = 0.58, *p* = .01). The pairs with high transition similarity also have high gaze similarity.

### 5.5. Game Phase 

**Gaze similarity and the second posttest score:** further, concerning the collaborative gaze patterns, we observe a significant correlation between the gaze similarity during the game phase and the second posttest score (r (16) = 0.45, *p* = .05). The pairs having high gaze similarity have high average second posttest score (Figure 12a). 

**Gaze similarity, speech and second posttest score****:** to study the relation between the gaze similarity, speech and the second posttest score, we examined the effect of the speech segments (speech and no speech) on the relationship between the second posttest score and the gaze similarity. Table 6 shows the ANCOVA results. We observe two single effects and an interaction effect. Both the second posttest score and the type of speech segments are related to the gaze similarity; and there is an interaction effect of these two variables on the gaze similarity. The second posttest score is correlated with the gaze similarity (F (1,16) = 5.54, *p* = .02, d = 1.10). We also observe that the gaze similarity is significantly higher for the speech segments than the gaze similarity for the no speech segments (F (1,16) = 12.26, *p* = .001, d = 1.65). Furthermore, the correlation between the gaze similarity and the average second posttest score is higher for the speech segments than that for the no speech segments (F (1,16) = 11.96, *p* = .001, d = 1.63 Figure 12b). 

**Competition vs. collaboration score****:** we did not observe any significant correlation between the difference in scores of the two players and their average score in the collaborative version of the game (r (16) = -0.10, *p* = .66).

**Competition vs. collaboration gaze similarity:** we observed a significant difference in the gaze similarity of the two players for the two different versions of the game (F (1,16) = 4.79, *p* = .02, d = 1.02). The gaze similarity was significantly higher for the collaborative version of the game than the gaze similarity for the competitive version of the game.

**Time on task for each question:** we observed no significant change in time to answer each question (Table 7 and Figure 13), barring the question numbers 12–14; only a few people could attempt these questions and that too in a hurry because it was towards the end of the game. 

**Use of power ups:** most of the power ups were used towards the end. We observe a positive correlation with gaze similarity (r (16) = 0.53, *p* = .02, Figure 14a) and a positive correlation with score (r (16) = 0.47, *p* = .04, Figure 14b). However, we did not observe any relation with the individual type of power up (double xp, pause time, hint). 

### 5.6. Poster Versus Game Phase 

For all the comparisons in this subsection of results, the gaze similarity in the games is considered from the collaborative phase of the game. The main reason behind this decision is that there is no practical, empirical or theoretical basis of having any considerable gaze similarity in the competitive version of the game. This was also evident when we compared the gaze similarities during the competitive and collaborative versions of the game and found that the gaze similarity during the competitive version of the game is significantly less than the gaze similarity during the collaborative version of the game.

**Gaze similarity during posters and during the game:** we observe a significant correlation between the gaze similarity during the poster phase and the gaze similarity during the game phase (r (16) = 0.49, *p* < .04). The pairs having high gaze similarity during the poster session also have high gaze similarity during the game phase (Figure 15a). 

**Gaze similarity during the game and transition similarities during the posters:** we observe a significant correlation between the transition similarity during the poster phase and the gaze similarity during the game phase (r (16) = 0.55, *p* = .01). The pairs having high gaze similarity during the poster session also have high gaze similarity during the game phase (Figure 15b).

**Gaze similarity during the game and AOI transitions during the posters:** we observe a significant correlation between the image to text transitions during the poster phase and the gaze similarity during the game phase (r (16) = 0.56, *p* = .01). We also observed a significant correlation between the text to image transitions during the poster phase and the gaze similarity during the game phase (r (16) = 0.50, *p* = .03).

## 6. Discussion and Conclusions 

We presented a dual eye-tracking study in the context of an informal collaborative setting. Dyads of participants were asked to explore and discuss the material provided in a “museum-like” setup. They were then required to play a gamified quiz. In this contribution, we used automatic gaze and speech analyses to explain the collaborative learning mechanisms (CLM) involved in the informal learning scenarios. 

From the results of the present DUET study we can explain the following:Using the DUET data one can understand both the individual and collaborative learning processes. For example, we observe that those individuals who understood the relation between the text and the visualizations correctly (high number of image to text and back transitions) had higher learning gains. Moreover, we also observe that those dyads who put efforts into establishing the common ground (looking at the same thing at the same time while talking about it) learned more (high posttest and game scores).We also provided an analysis that combined different tasks and showed that it is important to consider the behavior in both the interaction media (physical posters and digital games) to understand the successful learning processes. For example, we observe that the gaze similarity in the poster and game phases, and the scores (posttests and game) are correlated to each other.Finally, we empirically show how collaborative learning mechanisms could be understood using DUET data. For example, coordination of collaboration, joint attention and narration, can be captured using the overall gaze similarity and the similarity measures during speech episodes. Furthermore, collaborative discussion can be measured using the transition similarity during the speech episodes.

In this section, we elaborate on our contributions and provide detailed implications of our findings.

Regarding the game and test scores, the results show no significant improvement from the first posttest score to the second posttest score. Looking at the means and the standard deviation of the first (mean = 0.70, sd = 0.24) and the second (mean = 0.72, sd = 0.23) posttest, one can reason that there was a very small room for improvement for the students in the second posttest. However, there is a slight but nonsignificant improvement. The results also indicate towards a positively significant correlation between the two posttest scores, which shows that the interactive multi-touch display does not obstruct the learning process of the teams. This is similar to the results reported in the prior works done by Sluis et al. [37] and Zaharias et al. [6]. These two studies also report although the interactive displays do not significantly increase the learning gain, they also do not hinder them. Concerning the game score and the learning gain of students, the results show a positive correlation between the two quantities. Sangin et al. [75], in a collaborative concept map task, also reported a strong, positive and significant correlation between the learning gains and task-based performance of a team. 

The results, based on the eye-tracking data, presented in the previous section represent two behavioral gaze patterns (individual AOI transitions and collaborative gaze/transitions similarity), both of which are correlated to participants’ learning outcomes. 

Concerning the individual gaze patterns reported in this study, the gaze transitions between images and the corresponding text pieces play an important role in explaining the posttest scores. The results show that the students with a higher first posttest scores connect the text and images/graphics in a better manner and in higher proportions than the students with the low first posttest scores. The complementarity of the information presented in the textual form and the images is of utmost importance when it comes to understanding a learning material (as suggested by Meyer, [76]). By connecting the correct pieces of text to the appropriate image, the students understand the content more and hence obtain a high first posttest score. On the other hand, students who fail to do so end up with a low first posttest score. In two separate eye-tracking studies, Sharma et al. [50] and Strobel et al. [77] have shown that the understanding the content using the complementary information present in the different formats was a key process while learning with video lectures [8,50] or printed content [77]. 

Concerning the collaborative gaze patterns reported in this study, the gaze similarity between the peers depicts the amount of time spent by the partners looking together at the similar sets of objects in their field of view. This contribution measures the joint attention of the partners using gaze similarity. Joint attention is one of the key aspects of creating and sustaining high levels of mutual understanding. The results show a significant and positive correlation between the obtained knowledge and the gaze similarity of the pair. One plausible reason for this correlation could be the fact that while looking at the similar objects the peers might discuss and reflect upon the content of the posters which in turn sustains a high level of mutual understanding of the team. On the other hand, this process might be difficult if the peers are not looking at the similar set of objects together. Such results were also reported in different educational contexts [5,10,50,51,53,59]. These contributions have shown that the gaze cross-recurrence (or gaze similarity) is correlated to the learning outcomes and/or task-based performance in collaborative settings such as, collaborative programming [10,78], tangible interaction [5,51], concept maps [50], listening comprehension [53,54]. We found that the gaze similarity during the collaborative phase of the game to be significantly higher than the gaze similarity during the competitive phase of the game. This verifies and quantifies findings coming from relevant qualitative studies [63,64] which argue that collaboration promotes different behavior from competition, demonstrating a trait-like individual difference for students.

We also compared the gaze similarity during the collaborative and the competitive phases of the game. The results show that the gaze similarity during the collaborative phase of the game is significantly higher than the gaze similarity during the competitive phase of the game. On the other hand, the scores in the two phases were not significantly different. The difference in the collaborative gaze behavior during two game phases is driven by the need of the respective phases. For example, during the collaborative phase the peers would need to discuss the answer before answering the question and this would lead them to point to the different parts of the questions and the multiple choices available to them. This would in turn increase the gaze-similarity (as our results show that during speech episodes the gaze-similarity increases). On the other hand, during the competitive phase no such dialogue is necessary among the peers and most of them answered the questions quietly, this could also explain the lower values of the gaze-similarity. One might argue that in the competitive phase the gaze similarity should be zero. However, this would not be the case because the questions were made available to the peers at the same time hence, they would read part (or full) question at the same time. This would result in a non-zero value gaze similarity (still significantly less than the gaze-similarity during the collaborative phase) during the competitive phase.

We also computed the transitions similarity between the peers in a dyad. Transition similarity is an extension of the gaze similarity. Gaze similarity captures if the two peers are looking at the similar sets of the objects in their respective field of view in a given time window. On the other hand, transition similarity denotes that the peers are also shifting their attention among the different sets of objects in a similar manner. In other words, gaze similarity is the similarity in attention distribution, while the transition similarity is the similarity in the attention changes. We found that the transition similarity is higher for the pairs with high posttest scores than that for the pairs with low posttest scores. This shows that the pairs with high posttest score did not only focused on the similar object, but they also had similar change of the focus. 

Furthermore, this paper also presents the relationship between the collaborative gaze patterns in the two very different phases of the experiment, poster phase that was physical and the game phase that was digital. Another difference between the two phases was rooted in the script of the experiment. In the poster/physical phase of the experiment the students were not forced/told/asked to collaborate, they were simply told to “watch the posters as if they were in a museum with their friend”. This made the collaboration during the poster phase a voluntary effort of the students. On the other hand, during the game/digital phase of the experiment, according to the script of the experiment, the participants collaborated. Despite these two major differences, the gaze similarities in the poster/physical and the game/digital phases were significantly and positively correlated. This result indicates towards the validation of the interaction styles hypothesis by Sharma et al. [50], stating that in different collaborative settings the partners interact in two primary ways: “Looking AT” and “Looking THROUGH”. While Looking AT (teams with low gaze similarity in poster and game phases), the partners use the content as the main interaction point and while Looking THROUGH (teams with high gaze similarity in poster and game phases), the partners use the content as the medium to communicate and ground the conversation.

Finally, while comparing the gaze patterns across the different speech segments, we observe that the speech episodes have higher gaze and transition similarity then the no-speech episodes and this difference is wider for the participants with the high posttest score. In addition, this difference exists in both the poster and the game phases. This shows that there is higher amount of joint attention (measured by gaze similarity) and attention shift (measured by transition similarity) accompanying the verbal actions of peers in a dyad. Additionally, the gaze similarity is higher during the moments when the participants were using the power ups in the collaborative version of the game. This depicts that beneficial coordinated actions and joint attention episodes occur together. 

The correlation between the average posttest score for the dyads and the measures of their joint attention shows that there is a convergence of conceptual change. These results also show that CLM can be used in an automatic manner, although up to an extent. For example, the coordination of collaboration, joint attention and narration, can be captured using the overall gaze similarity and the similarity measures during speech episodes. The triumvirate relation between the posttest score, gaze similarity and the type of speech episodes indicates that there was a process responsible for creating and sustaining the mutual understanding trough references. Previous research has shown a close relation between the mutual understanding and the joint attention [10,79]).

On the other hand, the collaborative discussion can be measured using the transition similarity during the speech episodes. Transition similarity indicates a deeper level of mutual understanding as it does not merely capture the attention similarity but the similarity between peers’ shift of attention. This might be a result of more engaging dialogue process then simple deictic references since the high levels of transition similarity co-occur more with the speech episodes than the no-speech episodes. 

We used a simple script for the present study, which consisted of two phases based on the framework presented by Leftheriotis et al. [40]. In the first exploration phase, the dyads were asked to go through the learning material in a “museum-like” setting, and the second application phase was the game. The application phase had two different substages, a collaboration stage and a competitive stage. Competitive stage was the same game as the collaborative stage but with the peers playing against each other. The results (learning outcomes, game performance) show no differences in terms of which version of the game was first played by the team, this depicts the fact that both the collaboration and competition entail similar effect on the engagement of the users. This is reflected by the fact that most of the users indicated high levels of fun (mean = 6.4, sd = 0.87, 7-point Likert-scale), excitement (mean = 6.1, sd = 1.04, 7-point Likert-scale) and enjoyment (mean = 6.2, sd = 0.98, 7-point Likert-scale). The users also indicated that having a multi-touch display in such a setting is practical (mean = 6.0, sd = 0.97, 7-point Likert-scale). While the pairs were going through the posters, we observed significant discussion among the team members about the content of the poster. This might reflect the effectiveness of the simple script employed for the study as previously suggested by Giannakos et al. [27]. 

In this contribution, we showed that there are individual and collaborative gaze patterns, which can explain the learning outcomes/processes of the participants in a collaborative informal educational setting using automatic methods in a museum-like setting and through the utilization of interactive surfaces. These explanations are coherent with studies conducted in more formal educational settings and other in-the-wild studies that employ interactive surfaces and collect data utilizing other techniques (e.g., observations, interviews, movement tracking; [43,62]). However, the utilization of multi-modal learning analytics (in our case gaze and speech) in understanding the learning processes and outcomes around an interactive surface and comparison of those processes with a typical informal learning setting, allows us to extract insights about students’ collaboration and learning in an automated manner. This contributes to the contemporary body of research on the use of interactive displays to support learning and collaboration [1,44] and opens new research avenues. For example, how to incorporate this automated information to the interface and functionalities of interactive displays. Further development of these methods might have the same rigor as the qualitative analyses of videos and dialogues, which are time taking and cumbersome processes. Moreover, these results will also lead us to design more hands-on activities with interactive displays within the informal settings to study their influence on the learning outcomes. 

The present study is one of the first to use eye-tracking data to examine the relation between gaze, collaboration and learning gain in an informal learning scenario. However, our study entails some limitations. Mobile eye tracking devices are fragile, need frequent recharging and of course demand good calibration and thus additional time for the experiment was needed; in addition, the equipment is expensive. Due to the above limitations, there was a relatively small sample size, the total number of participants was 36 and most of them were males. With a small sample size, there is less statistical power. However, in most metrics the differences we found are significantly important. 

Moreover, the participants were volunteers from our university which may somewhat limit the generalizability of our results, so other sampling methods could have been applied to ensure a more consistent sample was obtained in terms of the age and reading skills. In addition, the experiment took place in a lab in the university and not in a museum or a similar real-life setting. Finally, this study lacked structured qualitative data (e.g., observations and interviews), which would be a fruitful opportunity for further research. 

Future studies could compare the role of gaze in alternative learning environments, as well as obtaining deeper insights from longitudinal collection of eye-tracking data. In addition, instead of posters with information, we are in discussions to run a similar study in a museum with real artefacts where participants could also interact with them and thus understand the role of gaze while interacting collaboratively in a real museum setting. 

## Figures and Tables

**Figure 1 sensors-20-01964-f001:**
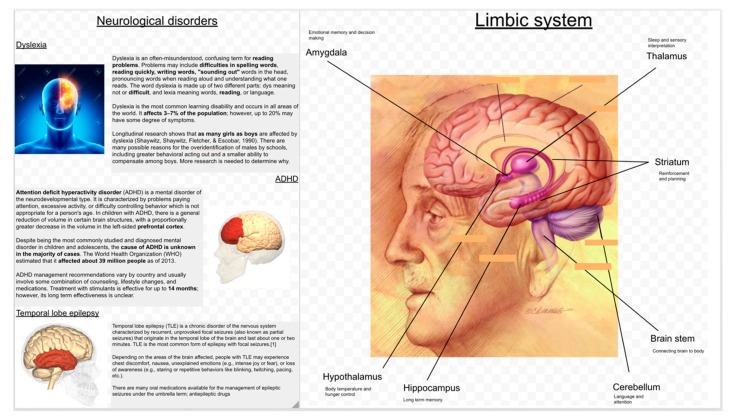
Examples of the posters, neurological disorders, and the limbic system.

**Figure 2 sensors-20-01964-f002:**
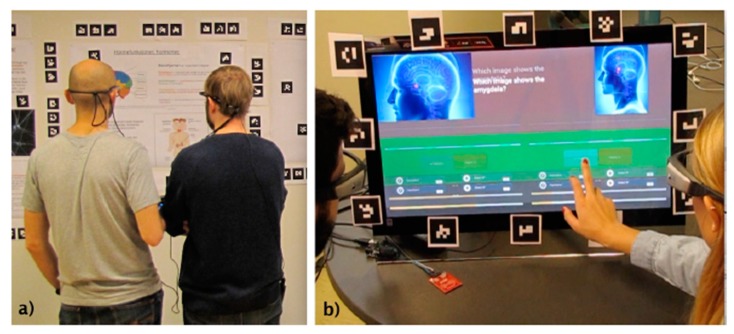
Experiment screenshots, (**a**) teams are watching the posters as they would do in a museum, and (**b**) teams are playing gamified quizzes (collaborative/competitive) on interactive display.

**Figure 3 sensors-20-01964-f003:**
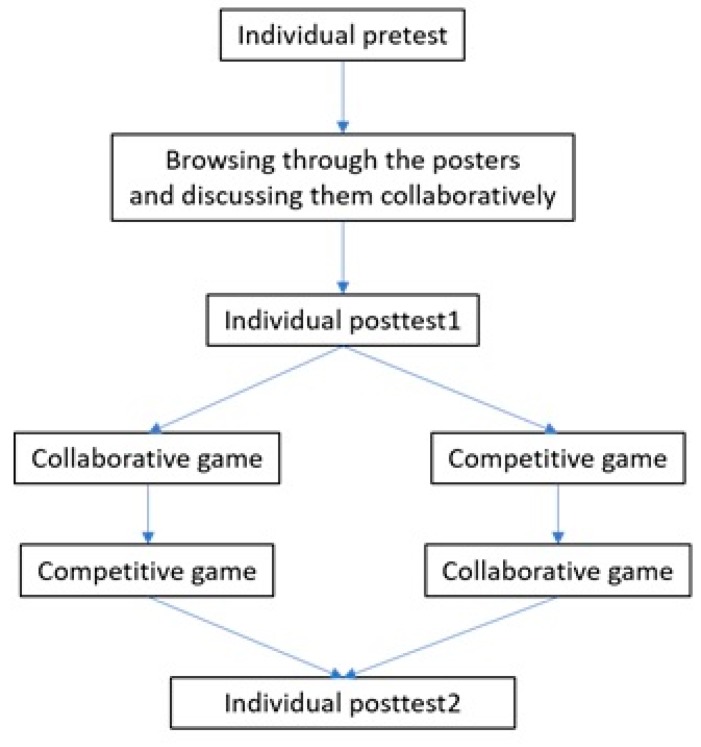
Different phases of the study.

**Figure 4 sensors-20-01964-f004:**
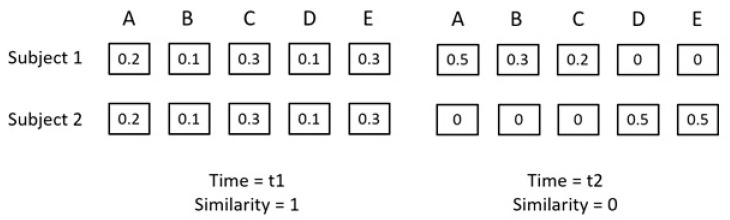
A typical example showing the situations for the lower and the upper limits of gaze similarity.

**Figure 5 sensors-20-01964-f005:**
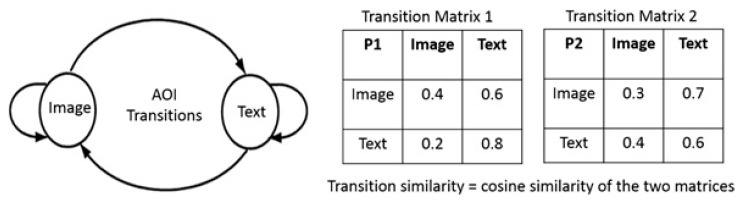
A typical example showing the situations for computation of gaze transition similarity.

**Figure 6 sensors-20-01964-f006:**
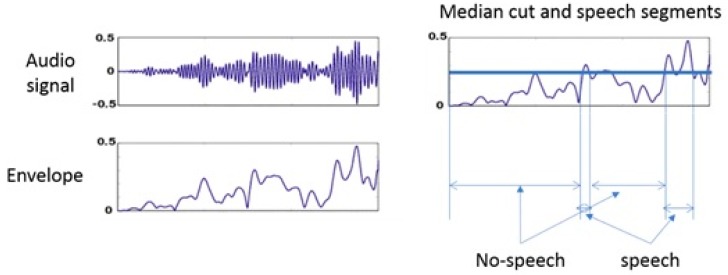
A typical example showing the computation of speech episodes.

**Figure 7 sensors-20-01964-f007:**
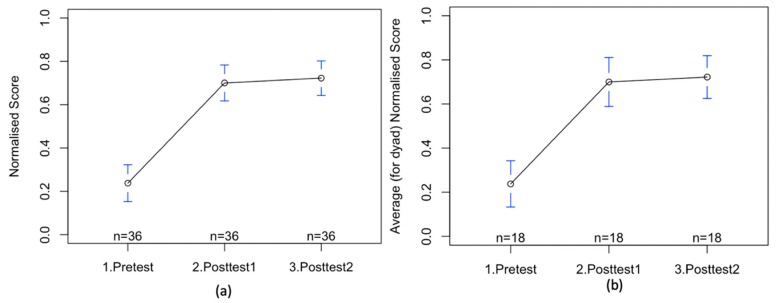
Comparison of scores from the pretest, the first and the second posttest. (**a**) for individuals; (**b**) for groups. All values are normalized between 0 and 1. The points show the mean values among all the participants and the blue bars show the 95% confidence intervals.

**Figure 8 sensors-20-01964-f008:**
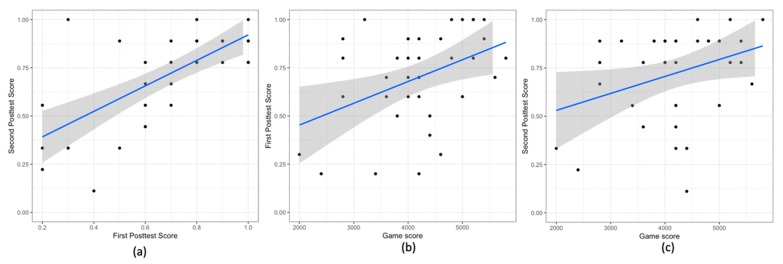
Scatter Plots for (**a**) first and second posttest score, (**b**) game score and the first posttest score, and (**c**) game score and the second posttest score. In all the plots the blue line shows the linear model for the variable on y-axis given the variable on the x-axis. The grey area shows the 95% confidence interval.

**Figure 9 sensors-20-01964-f009:**
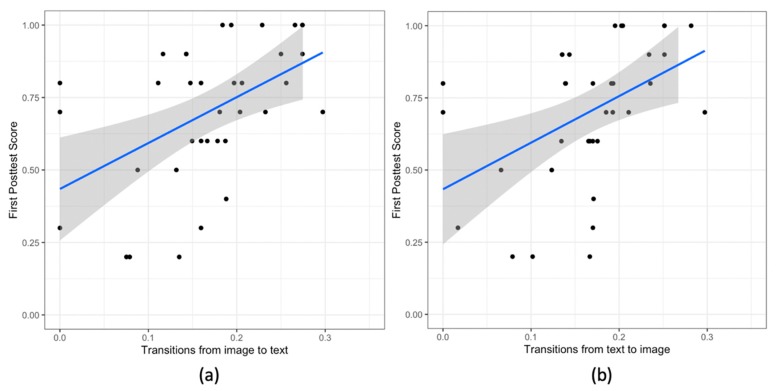
Scatter Plots for the individual transitions (**a**) image to text, and (**b**) text to image; and the score in the first posttest. In all the plots the blue line shows the linear model for the variable on y-axis given the variable on the x-axis. The grey area shows the 95% confidence interval.

**Figure 10 sensors-20-01964-f010:**
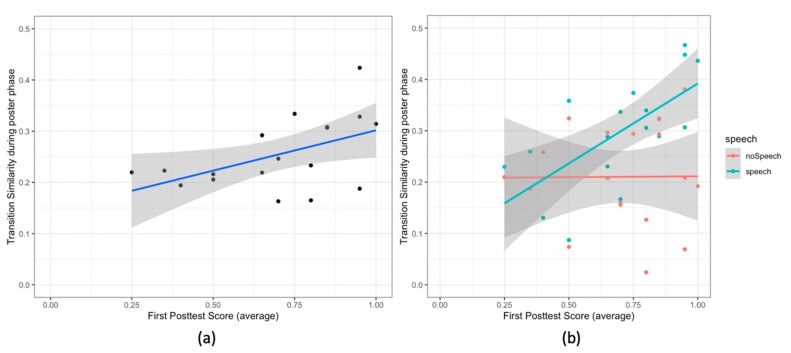
Scatter Plots for (**a**) the transition similarity and the score in the first posttest; (**b**) the same graph, but the colors show different speech segments (speech vs. no speech). In all the plots the lines show the linear model for the variable on y-axis given the variable on the x-axis. The grey area shows the 95% confidence interval.

**Figure 11 sensors-20-01964-f011:**
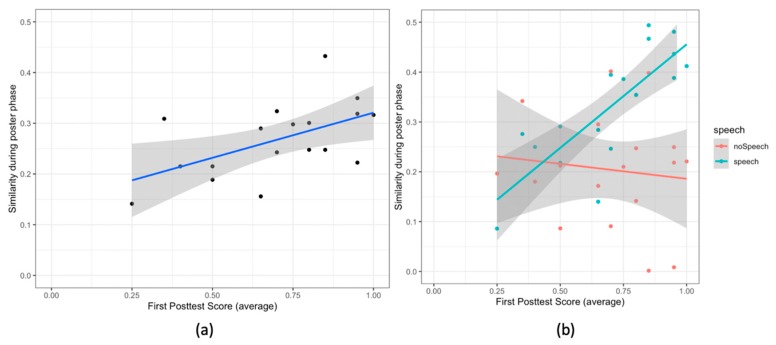
Scatter Plots for (**a**) the gaze similarity and the score in the first posttest; (**b**) the same graph, but the colors show different speech segments (speech vs. no speech). In all the plots the lines show the linear model for the variable on y-axis given the variable on the x-axis. The grey area shows the 95% confidence interval.

**Figure 12 sensors-20-01964-f012:**
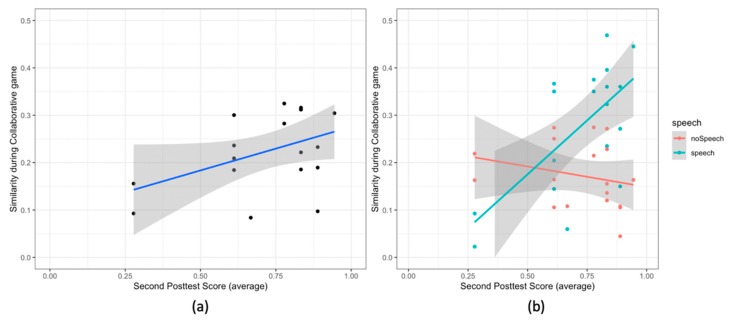
Scatter Plots for (**a**) the gaze similarity and the score in the second posttest; (**b**) the same graph, but the colors show different speech segments (speech vs. no speech). In all the plots the lines show the linear model for the variable on y-axis given the variable on the x-axis. The grey area shows the 95% confidence interval.

**Figure 13 sensors-20-01964-f013:**
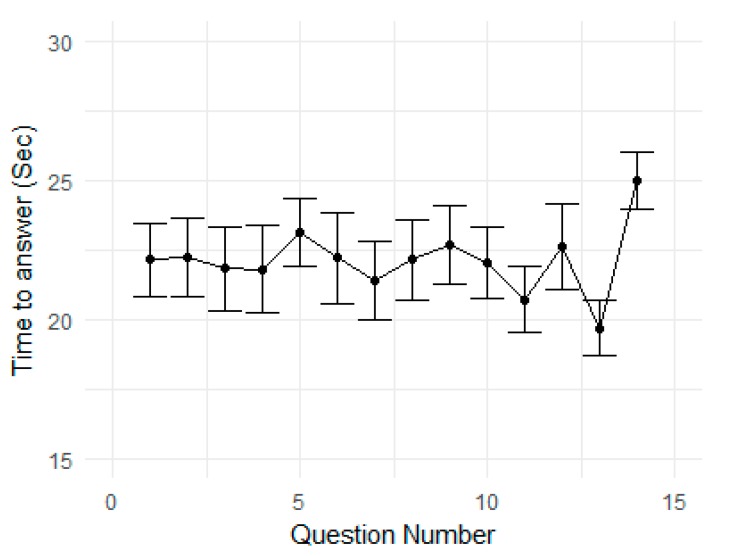
Mean time to answer each question in the game. The vertical bars show the 95% confidence interval.

**Figure 14 sensors-20-01964-f014:**
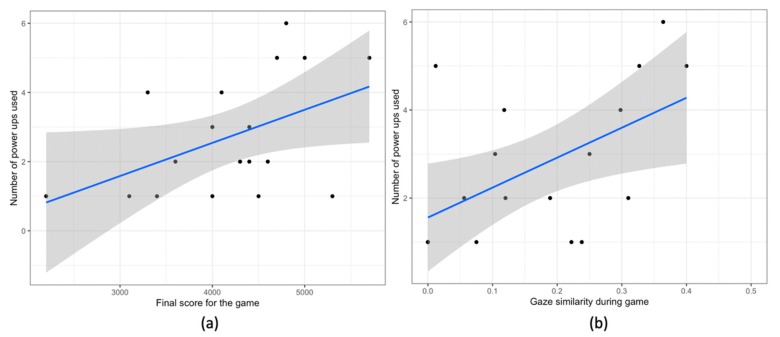
Scatter Plots for (**a**) the final game score and number of power ups used, and (**b**) collaborative gaze similarity during the game phase and number of power ups used. In all the plots the blue line shows the linear model for the variable on y-axis given the variable on the x-axis. The grey area shows the 95% confidence interval.

**Figure 15 sensors-20-01964-f015:**
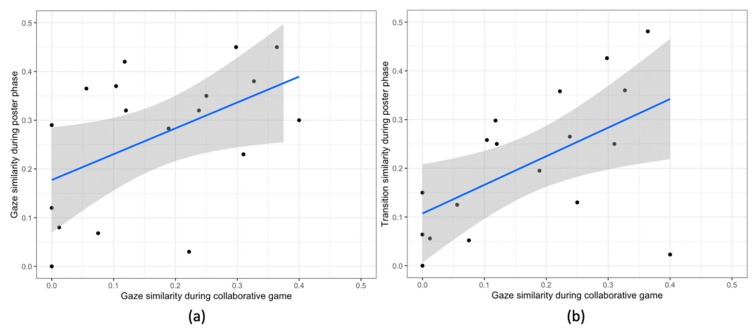
Scatter Plots for (**a**) the final gaze similarity during the poster and game phases, and (**b**) collaborative gaze similarity during the game phase and transition similarity during the poster phase. In all the plots the blue line shows the linear model for the variable on y-axis given the variable on the x-axis. The grey area shows the 95% confidence interval.

**Table 1 sensors-20-01964-t001:** Summary of design elements for the research space.

Design Rationale	Design Element
Easy and understandable structure [27]	Simple rules for the game and straightforward script for the task
Visual and interactive element [27]	Game elements like the avatars, authentication PICC.
Entertaining responses for on -screen answers [27]	Animal avatars, power ups
Interaction enhancement with large touchscreens [27]	Multi touch screens (size)
Direct attention to specific topics [27]	Individual questions to address the specific neuroscience phenomenon
Cooperation and competition among users [27]	Two different within subject conditions (order balanced)
Type of reward that user seem to be motivated to gain [62]	Double XP, pause time, Hint
Table tops for informal learning aids collaboration [62]	Multi touch vertical screens
Time-out feature to complement the mid game frustration due to difficulty [62]	Pause time and hints
Provide external trigger to prompt user engagement [62]	Every time users got a power up, they were shown the benefits of using them

**Table 2 sensors-20-01964-t002:** Descriptive statistics for the variables used in this contribution.

Variable	Mean	Std. Dev.	Minimum	Maximum
Pretest Score	0.24	0.25	0.00	0.10
First posttest Score	0.70	0.24	0.20	1.00
Second posttest Score	0.72	0.23	0.11	1.00
Image to text transitions	0.15	0.04	0.07	0.25
Test to image transitions	0.15	0.06	0.01	0.25
Transition similarity poster phase	0.20	0.14	0.00	0.48
Gaze similarity poster phase	0.27	0.13	0.00	0.45
Gaze similarity game phase	0.17	0.13	0.00	0.40
Game score	4188.88	845.67	2200	5700
Power ups used game phase	2.72	1.71	1	6

**Table 3 sensors-20-01964-t003:** Normality and Homoscedasticity test statistics for the tests and game scores.

	Shapiro–Wilk Test		Breusch–Pagan Test	
**Variable**	**W**	***p*-Value**	**BP**	***p*-Value**
**XP**	0.97	.44	0.64	.42
**Pretest**	0.85	.57	0.39	.32
**Posttest1**	0.90	.35	2.33	.12
**Posttest2**	0.85	.26	0.01	.91

**Table 4 sensors-20-01964-t004:** Interaction effect of the first posttest score and the type of speech segment on the transition similarity during the poster phase.

Model for the Transition Similarity	F-Value	*p*-Value	Effect Size
First post test score	5.01	.03	1.05
Speech segment	8.42	.006	8.42
Interaction term	4.79	.03	4.79

**Table 5 sensors-20-01964-t005:** Interaction effect of the first posttest score and the type of speech segment on the gaze similarity during the poster phase.

Model for the Gaze Similarity	F-Value	*p*-Value	Effect Size
First post test score	5.74	.02	1.12
Speech segment	15.60	.001	1.86
Interaction term	10.27	.003	1.51

**Table 6 sensors-20-01964-t006:** Interaction effect of the second posttest score and the type of speech segment on the transition similarity during the collaborative game phase.

Model for the Gaze Similarity	F-Value	*p*-Value	Effect Size
Second post test score	5.54	.02	1.10
Speech segment	12.26	.001	1.65
Interaction term	11.96	.001	1.63

**Table 7 sensors-20-01964-t007:** Pair-wise differences in time to answer each question. NA = not enough data for one of the questions in the pair.

Question pair	1-2	2-3	3-4	4-5	5-6	6-7	7-8
**t-value**	−0.08	0.39	0.02	−1.33	0.90	0.73	−0.72
**p-value**	.93	.69	.98	.18	.36	.47	.47
**Effect size**	0.02	0.13	0.01	0.45	0.30	0.25	0.24
**Question pair**	**8-9**	**9-10**	**10-11**	**11-12**	**12-13**	**13-14**	**14-15**
**t-value**	−0.49	0.62	1.39	−1.62	2.29	−2.95	NA
***p*-value**	.62	.53	.17	.11	.04	.03	NA
**Effect size**	0.16	0.21	0.47	0.55	0.78	0.01	NA

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
