# Peer review of "Utilizing Interactive Surfaces to Enhance Learning, Collaboration and Engagement: Insights from Learners’ Gaze and Speech"

_sensors, 2020, doi:10.3390/s20071964_

Round 1
Reviewer 1 Report
This paper explores the value of learner's gaze information in informal learning environments. Overall, this study is well-motivated and explained with backgrounds. The experiments are properly designed with details. The findings are interesting and supported by results.
There are however several parts to be clarified:
- Weak novel findings: In Discussion and Conclusion, this paper often mentions results by previous researches, but it makes the novel findings in this study weak.
- Unclear some terms: This paper uses words like "clicks" in Abstract and "actions" in Results as measurements, but there are no descriptions and results.
- Missing main claim: This paper mainly claims the term, "interactive surfaces" as a key issue, but there are not much findings and observations about it.
- Unclear the purpose of some comparisons and findings: (page 20) 1) "Gaze similarity during the game and transition similarities during the posters" and 2) "Gaze similarity during the game and AOI transitions during the posters"
- Page 18, line 588-590, Page 22, line 738-740: The results are interesting and need more explanation because in a natural sense, collaboration and competition are really different context, inducing different behaviors.
- Page 19 "Poster versus Game Phase": Did you consider both cases of the game phase (collaboration and competition) when comparing to the poster phase? As a result, for example, the pairs having high gaze similarity during the poster session might have high gaze similarity during the game phase in collaboration, but it is the same result when in competition?
Minors:
- page 11, line 407: is Figure 3 correct in the context of the text?
- page 9, line 364/its related part in Results: Please give some details about multiple-choice questions (what kind of questions, how difficult).
- Figure 7-12, Figure 14-15: The axis-information of plots is not consistent (min-max scale, interval, unit), so the plots are not visible in an intuitive manner. From the scatter plots, could we also easily say the observations have the "sufficient and obvious differences"?
- Page 21, line 648-650: is the citation correct in the context of the text ?
Overall, I would not argue that these are major flaws of this paper, but they should be clarified for the acceptance.
Reviewer 2 Report
This was an interesting read. I just have a couple of questions/comments:
1) There's need to be another round of English writing editing as I came across some minor error. For example, p.5 line 227 the "employees" should be "employs" or p.8 line 329 "has showed" should be "has shown".
2) p.7 line 316, what was the rationale for using 15 seconds pause time? Regardless of the type of gaming (i.e., collaborative/competitive), such a time constrain is contradictory to learning. This will change the assessment of learning to speed testing which is different than achievement testing and affects students' performance.
3) p.9 briefly describes the knowledge assessment plan but there is no information on the quality of quizzes. How did you determine content validity? what about the quality of items (e.g., difficulty, discrimination)? reliability of quizzes? These are essential for ensuring the quality of your learning data.
4) p.10 mentions creating random dyads and then states that the dyads did not necessarily stay together. This is a major error factor in the design. You need to explain why you put them into dyads and how this could affect your results. You also need to explain potential threats when dyads did not stay together and how you accounted for this.
5) p.11 lines 387-88, states that "they could go back to the posters and look for the answers.". Don't think this distorts your learning data? could this be one reason why you didn't see much gain in your third measurement? how do you interpret the effect of the time limit of 30 seconds on learning?
6) p.11 line 392-93, I don't understand this statement "We do not consider the learning gain in this experiment, as we observed a floor effect on the pretest scores (Mean = 0.2, Median = 0.1)". What do you mean by this?
7) p.12 Data Analysis section, this is the most confusing and vague part of the paper. I can't figure out what you did. You used counterbalancing of treatments, which is to account for order effect, so why are testing it?
7.1) You used "the first played phase as the independent variable for all the ANOVAs." what do you mean by that? did you it as covariate? Did you ran ANCOVA?
7.2) You said, "The independent variables were the game score, the pretest score, the first posttest score and the second posttest score." what do you mean by this? These are continuous variables and cannot be used as independent variables in an ANOVA model. Did you use Regression modeling?
7.3) p.14 line 492, "The scores in the first and the second posttests are correlated" Of course this would happen. This is a repeated measure design.
8) You have reported a couple of significant interaction effects. All significant interactions should be followed-up by simple effect analysis.
9) All significant effects (i.e. main or interaction) should be reported with appropriate effect size measures. Significance testing doesn't tell us much.
Reviewer 3 Report
sensors-75252
Title: Utilising interactive surfaces to enhance learning, collaboration and engagement: Insights from learners’ gaze
The authors present the results of a dual eye-tracking study utilising interactive surfaces to enhance learning, collaboration and engagement. A two-staged within-group experiment was conducted following single-group time series design, involving repeated measurement of participants’ gaze, voice, clicks and learning gain tests. The results show that collaboratively, pairs who have high gaze similarity have high learning outcomes.
Paper is well organized and the contributions are clearly stated.
To further improve this paper, the authors can consider the following suggestions:
- According to the Sensors template, the references should be numerically marked and organized in an ascending order.
- In methodology section, the authors should briefly describe the eye-tracking system used and how data was collected. Were there any limitations encountered using the dual eye-tracking system?
- How were the pre and posttest performed in order to evaluate the participants learning progress?
